# Effects of Surface Microstructures on Superhydrophobic Properties and Oil-Water Separation Efficiency

**Yangyang Chen** [1,2], **Shengke Yang** [1,2,*], **Qian Zhang** [1,2], **Dan Zhang** [1,2], **Chunyan Yang** [1,2], **Zongzhou Wang** [1,2], **Runze Wang** [1,2], **Rong Song** [1,2], **Wenke Wang** [1,2] and **Yaqian Zhao** [3]

[1] Key Laboratory of Subsurface Hydrology and Ecological Effects in Arid Region, Ministry of Education, Chang'an University, Xi'an 710054, China; 2015129085@chd.edu.cn (Y.C.); 2015129090@chd.edu.cn (Q.Z.); 2017129063@chd.edu.cn (D.Z.); 2017129066@chd.edu.cn (C.Y.); 2015229061@chd.edu.cn (Z.W.); 2016129077@chd.edu.cn (R.W.); 2016229036@chd.edu.cn (R.S.); 2017129064@chd.edu.cn (W.W.)
[2] School of Environmental Science and Engineering, Chang'an University, Xi'an 710054, Shaanxi, China
[3] Dooge Centre for Water Resource Research, School of Civil Engineering, University College Dublin, Belfield, Dublin 4, Ireland; 2016229046@chd.edu.cn
* Correspondence: yskfxh@chd.edu.cn or ysk110@126.com; Tel.: +86-29-8558-5589

**Abstract:** In order to explore the effects of microstructures of membranes on superhydrophobic properties, it is critical, though, challenging, to study microstructures with different morphologies. In this work, a combination of chemical etching and oxidation was used and some copper meshes were selected for grinding. Two superhydrophobic morphologies could be successfully prepared for oil-water separation: a parabolic morphology and a truncated cone morphology. The surface morphology, chemical composition, and wettability were characterized. The results indicated that the water contact angle and the advancing and receding contact angles of the parabolic morphology were 153.6°, 154.6° ± 1.1°, and 151.5° ± 1.8°, respectively. The water contact angle and the advancing and receding contact angles of the truncated cone morphology were 121.8°, 122.7° ± 1.6°, and 119.6° ± 2.7°, respectively. The separation efficiency of the parabolic morphology for different oil-water mixtures was 97.5%, 97.2%, and 91%. The separation efficiency of the truncated cone morphology was 93.2%, 92%, and 89%. In addition, the values of the deepest heights of pressure resistance of the parabolic and truncated cone morphologies were 21.4 cm of water and 19.6 cm of water, respectively. This shows that the parabolic morphology had good separation efficiency, pressure resistance, and superhydrophobic ability compared with the truncated cone morphology. It illustrates that microstructure is one of the main factors affecting superhydrophobic properties.

**Keywords:** superhydrophobic materials; rough morphology; parabolic morphology; truncated cone morphology; oil-water separation

## 1. Introduction

Both oil leakage accidents and arbitrary discharges of oily wastewater have caused great damage to water resources [1–4]. In order to recycle crude oil that has leaked into the water and to purify the water [5], a lot of efforts have been made to separate the oils from the water surface, including using oil skimmers, flotation, and gravity separation. However, most of them have some drawbacks, such as low separation efficiency, low flux, and high operation cost [6,7]. In recent years, inspired by surface structures like the lotus leaf self-cleaning surface and mosquito compound eyes [8], a series of superhydrophobic materials have been prepared and used for oil-water separation [9–12]; moreover, great advances have been achieved and this has attracted extensive attention from scholars.

Previous studies have shown that superhydrophobic properties are determined by surface energy; simultaneously, the water contact angle, and the advancing and receding contact angles are important

indexes to revealing superhydrophobic properties [13–18]. In order to represent superhydrophobic properties at a deeper level, S. Hoshian [19] and G. McHale [18] studied advancing and receding contact angles to better illustrate their importance and the superhydrophobic properties of membranes. Meanwhile, scholars continued to construct theoretical models in order to explore the theoretical relationship between all contact angles and superhydrophobic properties. The Wenzel model [20] was classical and it considered the rough surface to be grooved. However, it was different from the phenomenon of the actual superhydrophobic surface with lotus effect. The Wenzel model cannot explain the existence of droplets on a superhydrophobic solid surface. Yamamoto et al. [21] only thought that microscopic rough surfaces were pillar surfaces and did not consider the case where the protrusive surfaces were curved. Nosonovsky et al. [22] studied the relationship between surface roughness and the wetting properties of the sawtooth structures, periodic rough structures, rectangular structures, hemispherically topped cylindrical structures, conical structures, and pyramidal structures, and found that both hemispherically topped cylindrical structures and pyramidal structures can achieve a stable Cassie wetting state, which is consistent with the fact that the top surface of microscopic rough surfaces is often spherical or parabolic in the actual application. However, previous research work has lacked a more in-depth discussion of this situation. Eyal Bittoun and Abraham Marmur [23] constructed four different morphologies of rough surfaces: a cylinder, a truncated cone, a paraboloid and a hemisphere. They made a theoretical model analysis of wettability area for four different protrusive structural models in order to study the relationship between four different protrusive structural models and surface hydrophobicity. The conclusion was that parabolic protrusions seemed to be most advantageous, with the truncated cone second. However, the influence of morphology on surface hydrophobicity was only a theoretical prediction. It was necessary to verify the theoretical prediction results by fabricating superhydrophobic materials with different morphologies such as a truncated cone morphology and a parabolic rough morphology.

In addition, the correct choice of substrate materials is a key factor in achieving superhydrophobicity. Therefore, researchers have used different substrate materials: graphene sponge [24], poly vinylidene fluoride (PVDF) aerogel [25], zeolite [26], silicon [27], and copper [28], etc. to fabricate superhydrophobic surfaces. Superhydrophobic sponges can be prepared by the simple treatment of commercially available Pu-sponges with superhydrophobic nanoparticles [29]. Superhydrophobic surfaces can be created via surface modification of polymers [30]. However, the synthesis of graphene sponges involves multistep procedures, restricting their large-scale production for practical applications [31,32]. With PVDF aerogel it is difficult to remove the absorbed oil quickly, reducing the recyclability [33,34]. For absorbent materials like zeolite, it was difficult to recover oil, and material durability was poor, etc. [35]. The major drawback of silicon-based surfaces is mechanical fragility [19]. Copper meshes have attracted widespread attention for their mechanical strength, low density, high specific surface, and environmentally friendliness [36,37]. The as-prepared copper materials are promising for the preparation of superhydrophobic materials [38].

In addition, to fabricate superhydrophobic property surfaces, a rough structure must be modified with low surface energy coatings [39,40]. Methods reported include: chemical vapor deposition [41], hydrothermal [42], electro spinning [43], and etching [44]. However, many scholars have further improved the superhydrophobicity of the surfaces of copper meshes by etching and oxidation [38] but have not paid attention to variation in surface morphologies. It is significant to obtain superhydrophobic surfaces with different morphologies by etching and oxidation.

In recent years, further research has indicated that different pretreatment of the substrate material can obtain superhydrophobic materials with different morphologies and superior performance. Pan et al. [45] pretreated copper meshes with acetone, ultrapure water, HCl, nitric acid, methanol, etc. Maryam Khosravi et al. [46] pretreated stainless steel meshes with acetone, deionized water, $FeCl_3$, ethanol, carbon soot (CS), pyrrol liquid, etc. Rong et al. [25] pretreated copper foam meshes with diluted HCl, ethanol, deionized water, ammonium persulfate (($NH_4$)$_2$$S_2$$O_8$), dibasic sodium phosphate ($Na_2HPO_4$), etc. Elmira Velayi et al. [6] pretreated ZnO with deionized water, hexamethylenetetramine

(HMTA), ethanol, distilled water, etc. Although different materials and pretreatment methods had been used to prepare rough surfaces, the rough structures had not been described in detail and associated with theoretical models.

In this paper, copper meshes were used as substrate materials. The copper meshes were pretreated through a series of processes including etching, oxidation, grinding, and modification. A parabolic morphology and a truncated cone morphology were obtained. Finally, the two kinds of meshes' membranes were characterized respectively and applied to an oil-water separation. The hydrophobic properties of the superhydrophobic materials with different morphologies and their oil-water separation efficiency were verified.

## 2. Experimental

### 2.1. Materials

Copper meshes (200 meshes) were purchased from Shenyang Copper Network Co., Ltd. (Shenyang, China). Acetone (purity > 99%), ethanol (purity = 99.5%), benzene (purity > 99%), stearic acid (SA), carbon tetrachloride (purity > 99%), $FeCl_3$ (35 wt %) (purity > 98%) and $H_2O_2$ (30 wt %) were purchased from Tianjin Kemiou Chemical Reagent Co., Ltd (Tianjin, China). Engine oil was purchased from Hubei Jianyuan Chemical Co., Ltd (Wuhan, China). SiC papers (800 meshes) were purchased from Shanghai Chaowei Nano Technology Co., Ltd (Shanghai, China). The experimental water was deionized water. All other chemicals were analytical-grade reagents.

### 2.2. Fabrication of Superhydrophobic Materials

Copper meshes (200 meshes) that were cut into sizes of 4 cm × 4 cm were tilted into a beaker. The copper meshes were then cleaned ultrasonically in acetone, ethanol, and deionized water for 10 min to remove oil and inorganic impurities on their surfaces. The cleaned copper meshes that were placed flat on the configured 35% $FeCl_3$ solution were washed and etched in an ultrasonic bath for 20 min. The etched copper meshes were placed in 30% $H_2O_2$ solution and etched ultrasonically for 1 min. The as-prepared copper meshes surfaces were ground 50 times with SiC papers. The as-prepared copper meshes were then immersed in a 10 mol/L ethanolic solution of SA (with a volume ratio of 1:3, which refers to the volume ratio of ethanol to water) at 60 °C for 30 h. All experiments were performed at room temperature. Finally, the copper samples were successively rinsed with ethanol and deionized water and dried in air at room temperature and sealed for use. In order to better illustrate the preparation process of the two morphologies, the preparation process has been represented by a schematic diagram, as shown in Figure 1.

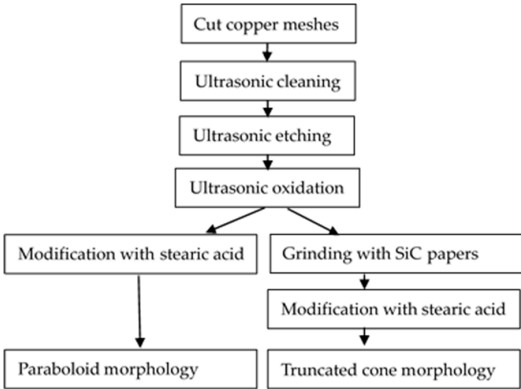

**Figure 1.** Preparation process schematic diagram of the two morphologies.

### 2.3. Oil-Water Separation Test

The as-prepared copper meshes that were trimmed into dimensions of 4 cm × 4 cm were fixed on a self-made separation apparatus. The oil/water separation performance of the copper meshes was determined by gravity-driven and capillary force-driven oil/water mixture separation experiments. An oil/water mixture of benzene-water, carbon tetrachloride-water, and engine oil-water was used for the separation experiments. For convenience, deionized water was stained with methylene blue. Oils were stained with Sudan I.

### 2.4. Characterization

#### 2.4.1. Surface Morphology Characterization

The surface morphologies of the SA-coated copper meshes obtained in different etching solutions were characterized using a microscope (LW300LFT, Beijing Taike Instrument Co., Ltd. Beijing, China) together with recording SEM imaging on a FEI Quanta 200 SEM (FEI company, Hillsboro, OR, USA) to confirm the successful formation of the microstructure. The chemical composition of the copper mesh surface modified with SA was analyzed via Fourier transform infrared spectroscopy (FT-IR) (Magna-IR 560, Nicolet, Beijing, China) and X-ray photoelectron spectroscopy (XPS) (AXIS ULTRA, Kratos Analytical Ltd., Kyoto, Japan)

#### 2.4.2. Contact Angle Measurements

The water contact angles (WCAs), oil contact angles, and advancing and receding contact angles of the samples were measured at ambient temperature using a SL200KS contact angle meter (American Kono Industrial Co. Ltd., Seattle, WA, USA); the volume of the test water droplets and oil droplets were approximately 3 μL. At least three parallel positions on the surface were measured to obtain average contact angle values. The advancing and receding contact angles were determined by tilting experiments.

## 3. Results and Discussion

### 3.1. Surface Microstructure after Ultrasonic Etching and Oxidation

Surfaces of the copper meshes with different roughnesses were obtained by etching with 35% $FeCl_3$ and oxidation with 30% $H_2O_2$ solution. In order to further observe the surface microstructure of the copper meshes, however, since the microscope could only roughly see the differences in surface morphology of the copper meshes, the copper meshes were characterized by SEM. Changes in the surface of the copper meshes were observed by microscope and SEM, respectively.

Figure 2 shows microscope and SEM images of the copper meshes before and after etching and oxidation. As shown in Figure 2a, the size of the copper meshes that were untreated was uniform and the thickness of the copper meshes was basically the same. Figure 2b shows that the diameter of part of the etched and oxidized copper mesh wires was slightly narrowed. Figure 2c,d are SEM images under 800× magnification. Comparing Figure 2c,d, it can be seen that the surface of the untreated copper meshes was smooth. The etched and oxidized copper meshes had no change substantially in shape, but the surface became rough. Moreover, the average sizes of the meshes shown in Figure 2c,d were determined as 71 and 70 μm, respectively, by measuring 10 times randomly. The diameter of the copper mesh wire was slightly narrowed. It was basically consistent with what was observed under the microscope, indicating that after ultrasonic etching using the acidic etchant $FeCl_3$ solution and ultrasonic oxidation using the $H_2O_2$ solution, the copper meshes were subjected to cavitation impact and the surfaces of copper substrates were deformed and etched, which confirmed that cavitation had occurred [45]. Moreover, the surfaces of the copper substrates became rough, which proved that the etching and oxidation processes increased the surface roughness to a certain extent [38,47,48].

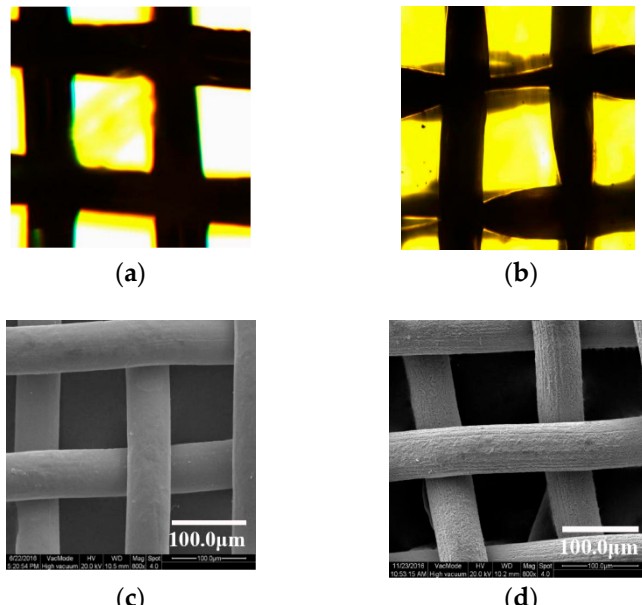

|  |  |
|:---:|:---:|
| (**a**) | (**b**) |
| (**c**) | (**d**) |

**Figure 2.** Copper mesh morphologies under microscope and SEM: (**a**,**c**) untreated; (**b**,**d**) etched and oxidized.

### 3.2. Surface Microstructure by Grinding with SiC

In order to change the surface morphologies of the copper meshes, several steps were performed by etching with 35% $FeCl_3$, using oxidation with 30% $H_2O_2$ solution, and grinding with SiC. Figure 3 shows microscope and SEM images of those etched and oxidized copper meshes that were ground with SiC and without SiC.

Comparing Figure 3a,b, there were some differences in the surface morphologies of the copper meshes before and after grinding with SiC. As in Figure 3a, the unground copper mesh surfaces were even. As in Figure 3b, the edges of the copper meshes that were ground with SiC were uneven and the surface became rough. It can be seen via microscope that the surface of copper meshes ground with SiC can become rough [49]. To deeply understand the surface morphologies of as-obtained samples before and after grinding, SEM analysis was conducted on copper meshes. Figure 3c,d are SEM images under 5000× magnification. In Figure 3c, the surface of the mastoid structure of the copper meshes was smooth and the papillae were regularly parabolic or granular protrusions. In Figure 3d, the papillae structures of the copper meshes had distinct angular structures and were arranged in a uniform and orderly manner. It was observed by SEM that the surface of parabolic or granular protrusions had been flattened after grinding by SiC, indicating that certain morphology features had appeared on the surface of the copper meshes before and after grinding.

|  |  |
|:---:|:---:|
| 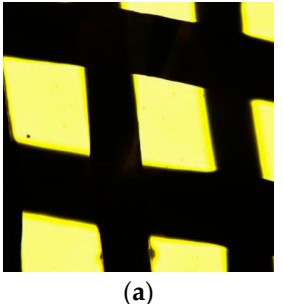 | 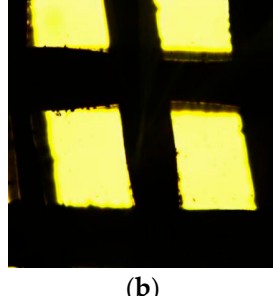 |
| (**a**) | (**b**) |

**Figure 3.** *Cont.*

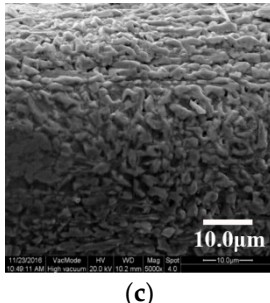
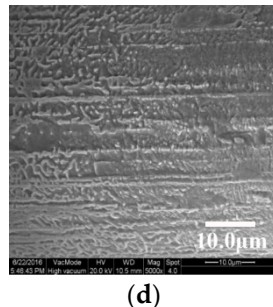

(**c**)                    (**d**)

**Figure 3.** Copper mesh morphology after etching and oxidation under microscope and SEM: (**a**,**c**) unground; (**b**,**d**) ground.

### 3.3. Surface Microstructure by Modification with SA

Previous studies have shown that using SA as modifier can form a superhydrophobic membrane surface on the surface of the materials [50–52]. Different microstructures can be obtained by etching with 35% FeCl$_3$, oxidation with 30% H$_2$O$_2$ solution, grinding with SiC, and modification with SA. Thus, the surface microstructures of the copper meshes that were modified with SA were observed via microscope and SEM.

Figure 4 shows the microscope and SEM images of the treated copper meshes that were modified with SA. In Figure 4a, it can be clearly seen that the surface morphologies of the copper meshes were still those of porous structures and had not been completely covered by SA, and only the pore size of the copper meshes had been reduced. In Figure 4b, a layer of SA on copper meshes was clearly evident.

SEM images were performed to further prove the successful formation of SA-decorated copper meshes. In Figure 4c, as can been seen from the fact that the surface structures of the copper meshes were comprised of many microprotrusions, and that many petallike clusters were present on the surface of the copper mesh surfaces, there are obvious spacings and gaps between the mastoid clusters which can capture a lot of air, demonstrating that air pockets were necessary to achieve surface superhydrophobicity [25,53,54]. The morphologies of many petallike papilla clusters were parabolic, arranged uniformly and with height kept consistent. Furthermore, air trapped under the space of protrusion clusters can cause the presence of air cushions on the surface of membranes which can prevent contact between water droplets and the solid surface, thereby increasing the contact angle of the water droplets [38,54]. Figure 4c shows that the copper meshes can form a rough structure similar to a paraboloid. In Figure 4d, a number of blocky and flaky bulges are seen to have arisen on the surface and the blocky mastoids formed by agglomeration of these protrusive structures appear as columnar, as if a series of rough structures with a uniform height and a truncated cone had appeared on the surface of the copper substrates. Figure 4d shows that the copper meshes can form a rough structure similar to a truncated cone.

In order to prove that the two morphologies obtained from Figure 4 are a paraboloid and a truncated cone, respectively, size fitting curves that are drawn by SEM images 4c,d are shown in Figure 4e,f. It can be clearly seen from Figure 4e,f that the surface morphologies of the unground copper meshes were similar to a paraboloid and the surface morphologies of the copper meshes that were ground with SiC were similar to a truncated cone.

Figure 5 shows the FT-IR pattern of the stearic acid and prepared hydrophobic surfaces, respectively. Figure 5a indicates the FT-IR pattern of stearic acid. The spectrum of stearic acid exhibits adsorption peaks at about 2917 and 2852 cm$^{-1}$ in the high frequency region, which, respectively, are attributed to C–H asymmetric and symmetric stretching vibrations. Moreover, due to the presence of carboxyl groups in stearic acid, a characteristic peak appears at 1707 cm$^{-1}$. The peak at 1707 cm$^{-1}$ represents the stretch absorption of the C=O bond. Figure 5b is the FT-IR pattern of a sample soaked with 10 mmol/L of stearic acid in an ethanol/water solution for 30 h. When the copper mesh was immersed in the stearic acid ethanol/water solution, the same two peaks as for stearic acid appeared

at 2917 and 2852 cm$^{-1}$. The surface of the modified copper mesh showed new characteristic peaks at 1720 and 1590 cm$^{-1}$. The two adsorption peaks may stem from asymmetric and symmetric stretches of carboxyl groups. Based on the analysis of the FT-IR spectrum, it can be seen that the surface of the copper meshes has been successfully modified with SA.

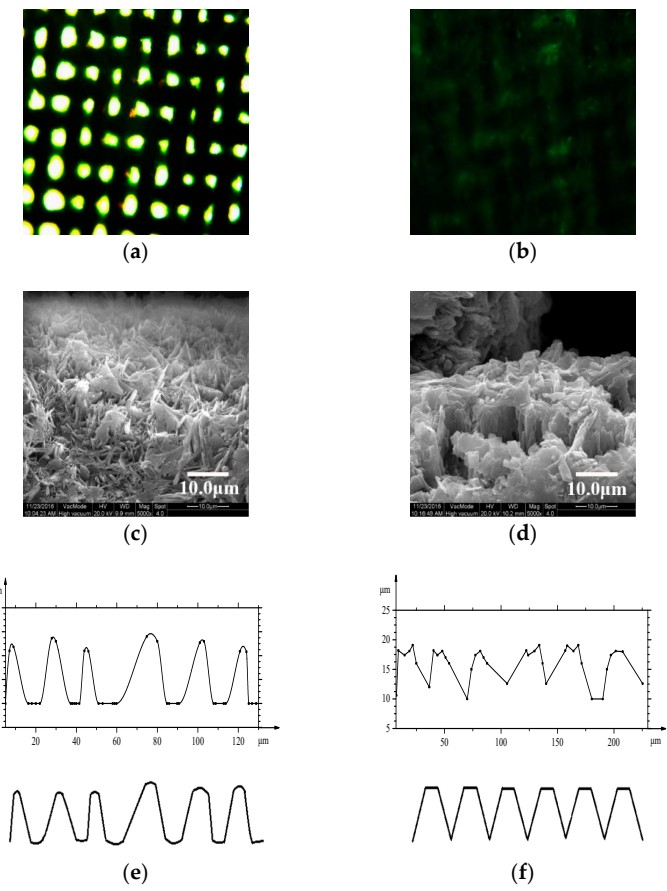

**Figure 4.** Copper mesh morphology after etching, oxidation, and modification with SA under microscope and SEM: (**a**,**c**) unground; (**b**,**d**) ground; (**e**) the fitting curve of Figure 4c; (**f**) the fitting curve of Figure 4c.

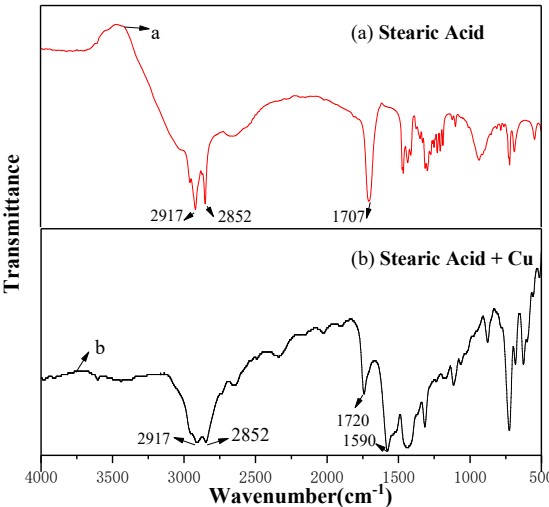

**Figure 5.** Fourier transform infrared spectroscopy (FT-IR) spectra: (**a**) stearic acid; (**b**) stearic acid + Cu.

To investigate the properties of superhydrophobic membranes, the surface composition was characterized by X-ray electron spectroscopy. Figure 6a shows the XPS survey spectrum of the as-prepared superhydrophobic copper surface. Figure 6b,c are high-resolution spectra of Cu 2*p* and C 1*s* respectively. As can be seen from Figure 6a, the peaks of oxygen, carbon and Cu appeared on the survey spectrum. The presence of C 1*s* and O 1*s* indicated the presence of SA in the membranes. However, the peak of Cu was not very obvious, probably because the coating of SA was thicker (100–300 μm), resulting in a weaker peak signal of Cu. It can be seen from Figure 6b that the main Cu $2p_{3/2}$ and $2p_{1/2}$ can be observed at the binding energies around 933.56 and 953.02 eV, respectively. As can be seen from Figure 6c, the C–C bond, the C–O bond, and the O=C–O bond appeared at 284.8, 285.63, and 289.11 eV, respectively. Moreover, a weak Cu–C bond was formed at 290.3 eV. Due to the presence of this weak bond, the copper meshes and SA were more closely linked together. It can be seen from Figure 6b,c that there was $Cu^{2+}$ on the Cu 2*p* peak and there was COO– on the C 1*s* peak; thus, these results indicate that SA was well coated on the copper mesh membrane.

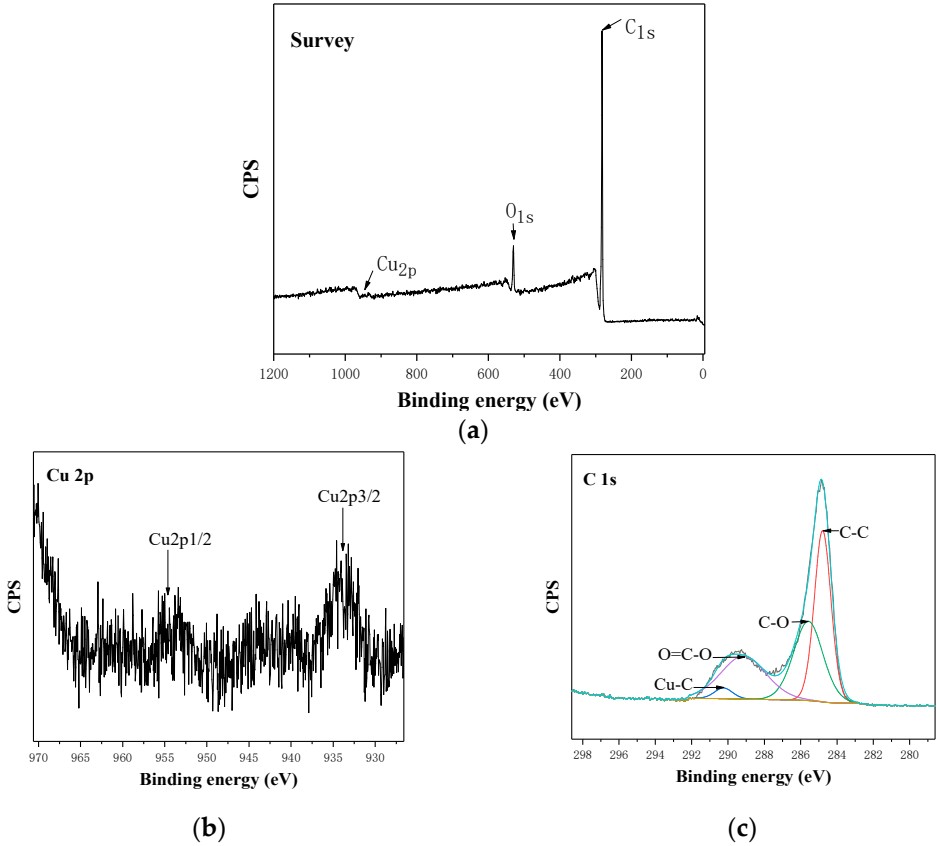

**Figure 6.** X-ray electron spectroscopy (XPS) spectra of superhydrophobic copper modified in an ethanol solution of SA for 30 h: (**a**) survey; (**b**) Cu 2*p* spectra; (**c**) C 1*s* spectra.

### 3.4. Wettability of the Copper Meshes

The water contact angle, advancing contact angle ($\theta_A$) and receding contact angle ($\theta_R$) are critical to the characterization of superhydrophobic surfaces. Figure 7 shows the WCA and the advancing and receding contact angles of the parabolic and truncated cone morphologies. In Figure 7a–c, the WCA and the advancing and receding contact angles of the parabolic morphology were 153.6°, 154.6° ± 1.1°, and 151.5° ± 1.8°, respectively, which showed that the parabolic morphology became a superhydrophobic surface after treatment with SA. In Figure 7b–d, the WCA and advancing and receding contact angles of the truncated cone morphology were 121.8°, 122.7° ± 1.6°, and 119.6°

$\pm$ 2.7°, respectively, which showed that the truncated cone morphology became a hydrophobic surface after treatment with SA. However, it was not a superhydrophobic surface.

Cao et al. [38] have synthesized copper meshes that were modified by n-dodecanethiol (DDT), and the surface of the copper meshes measured a static WCA of 152°, which showed that it was very important to grasp the surface structure and manufacturing process for superhydrophobic properties. Yuan et al. [55] synthesized copper meshes via a combination of polydimethylsiloxane (PDMS) template and etching method which produced a rough surface similar to a lotus leaf shape, upon which the surface of the copper meshes modified with SA measured a static WCA of 131°, demonstrating that it had a smaller contact angle than for the copper meshes etched and oxidized ultrasonically, which confirms that with ultrasonic etching and oxidation, the as-prepared meshes with nano-wall surface structures show good superhydrophobicity.

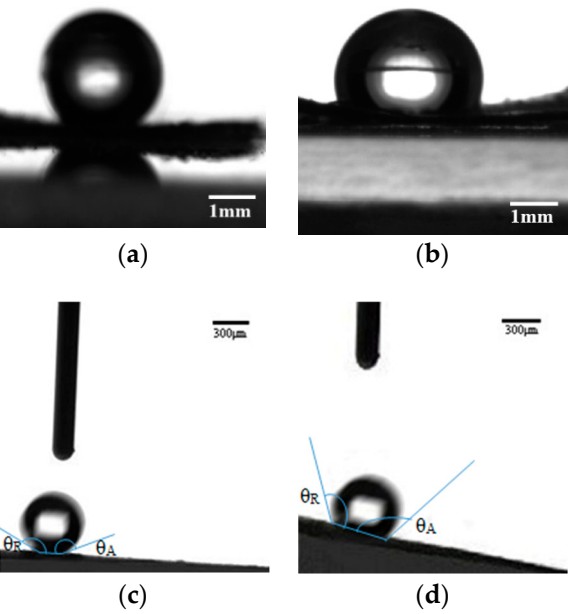

**Figure 7.** Wettability of the surfaces of the mesh membranes; water droplets on: (**a**,**c**) parabolic morphology; (**b**,**d**) truncated cone morphology.

The wetting properties of the as-prepared copper meshes towards oil (for engine oil and carbon tetrachloride, the surface tension of engine oil is 0.04 N/m and the surface tension of carbon tetrachloride is 0.02683 N/m [56])were evaluated through static contact angle. Figure 8a,b show the wettability images of engine oil on the paraboloid and truncated cone morphologies, respectively. Figure 8c,d show the wettability image of carbon tetrachloride on the paraboloid and truncated cone morphologies, respectively.

In Figure 8a, after three measurements being performed and averaged, the surface of the copper meshes produced a CA of 5°, which showed that the structure of the paraboloid became a superoleophilic surface. In Figure 8b, the surface of the copper meshes produced a CA of 32.9°, which showed that the structure of the truncated cone became a lipophilic surface. However, it was not a superoleophilic surface. Figure 8c,d shows that the viscosity of carbon tetrachloride was lower than engine oil and thus, the carbon tetrachloride could easily spread onto the copper mesh surfaces as carbon tetrachloride dropped on the copper meshes. The surface of the copper meshes produced a CA of 0.1°, which indicates that the microstructures of the paraboloid and truncated cone were superoleophilic. By comparing Figure 8a,c, and Figure 8b,d, it can be recognized that when the oil types were different (for example, engine oil/carbon tetrachloride), the contact angles on the surface of the material were different.

The wettability of water droplets, engine oil, and carbon tetrachloride on the paraboloid and truncated cone was measured, confirming that the microstructures of the surfaces had great influence on the CA and hydrophobic and lipophilic properties of materials [57,58]. Moreover, the different types of oil also led to differences in the CA on the material's surface. In summary, the rough structure of the paraboloid was superhydrophobic and superoleophilic, demonstrating it was more beneficial to achieve superhydrophobicity and superoleophilicity. The rough structure of the truncated cone was hydrophobic and lipophilic; however, it was not a superhydrophobic surface, which confirms that special surface morphologies are important factors that have great influence on surface superhydrophobicity [59,60].

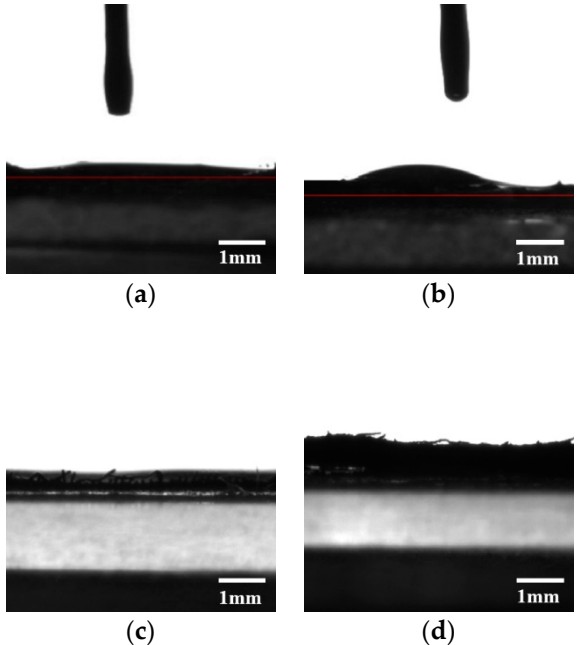

**Figure 8.** Wettability of the surfaces of the mesh membranes; oil droplets on: (**a**,**c**) parabolic morphology; (**b**,**d**) truncated cone morphology.

To better illustrate the contact properties of water droplets and various oil droplets on the two morphologies, all the contact angles are listed in Table 1.

**Table 1.** Contact angles of different materials on different morphologies.

| Different Situations | Different Types of Contact Angles | Angel | |
| --- | --- | --- | --- |
| | | Parabolic | Truncated Cone |
| water | WCA | 153.6° | 121.8° |
| | $\theta_A$ | 154.6° ± 1.1° | 151.5° ± 1.8° |
| | $\theta_R$ | 122.7° ± 1.6° | 119.6° ± 2.7° |
| Engine oil | – | 5° | 32.9° |
| Carbon tetrachloride | – | 0.1° | 0.1° |

## 3.5. Oil-Water Separation Performance

In order to understand the oil-water separation performance of the two morphologies, parabolic morphology was taken as an example. Carbon tetrachloride-water and benzene-water were selected for oil-water separation experiments. Deionized water was dyed with methylene blue, and carbon tetrachloride and benzene were dyed with Sudan I. The volumes of deionized water, engine oil and carbon tetrachloride used for the experiments were both 30 mL. The volume of benzene was 45 mL.

The separation experimental processes of benzene-water are shown in Figure 9a. The density of benzene was less than water and benzene passed first through the top glass tube and then flowed into the segregation apparatus. Due to the hydrophobic and lipophilic properties of the membranes, benzene passed through the surface of the membranes and flowed into the beaker. The deionized water was repelled on the membranes. The entire separation process took about 2 min. Engine oil underwent the same process as benzene.

The separation experimental processes of carbon tetrachloride-water are shown in Figure 9b. The density of carbon tetrachloride was greater than water and the deionized water passed first through the top glass tube and then flowed into the segregation apparatus. Due to the hydrophobic and lipophilic properties of the membranes, the deionized water was repelled by the membranes while carbon tetrachloride passed through the deionized water, reached the surface of the membranes, and flowed into the beaker quickly. The entire separation process took about 2 min.

When the separation process was completed, the oils in the beaker were poured into a measuring cylinder and the volumes of oils were measured after separation. It was found that the volumes of benzene, carbon tetrachloride and engine oil after separation using the parabolic morphology were approximately 43.8, 29.2, and 27.3 mL, respectively. The volumes of benzene, carbon tetrachloride, and engine oil after separation via the truncated cone morphology were approximately 41.7, 27.6, and 26.7 mL, respectively.

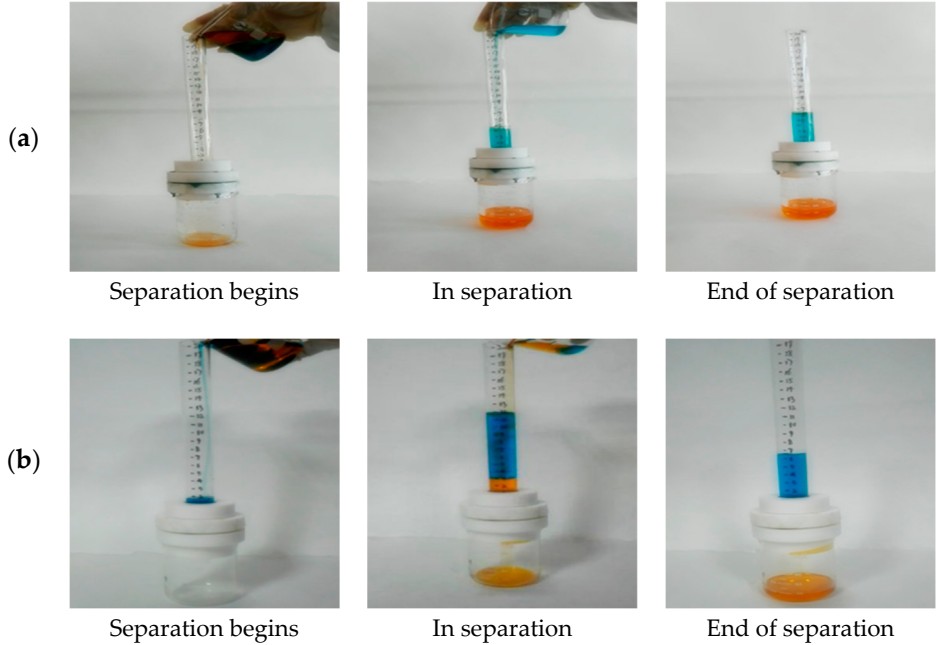

**Figure 9.** Separation effect diagram of oil-water mixture using parabolic membranes: (**a**) benzene-water; (**b**) carbon tetrachloride-water.

According to the above method a mixture of benzene, carbon tetrachloride, engine oil, and deionized water was supplied to oil-water separation experiments on the surface of the truncated cone. In order to assess the quality of separation of two different microstructures of the copper meshes, after separation of oil and water, according to the amount of oil passing through the filter, the separation efficiency, η (%), was calculated using Equation (1), which is

$$\eta = V_R/V_0 \times 100\% \tag{1}$$

where $V_0$ is the oil mass of the original oil-water mixture and $V_R$ is the oil mass after separation. The results obtained are shown in Figure 10.

It can be seen that the separation efficiency of the as-prepared rough morphology of the paraboloid was above 91% for three different oil-water mixtures of benzene, carbon tetrachloride, and engine oil. What was more important was that the separation efficiency of benzene and carbon tetrachloride was about 97%. However, given that the separation efficiency of the mixtures of benzene-water, carbon tetrachloride-water and engine oil-water was 93.2%, 92%, and 89% on truncated cone morphologies, it can be found that the oil-water separation efficiency of the rough morphology of the paraboloid was higher than for the truncated cone.

It can be recognized that due to the different types of oil, the contact angles of oil droplets on the surfaces of the copper meshes were different, so the separation efficiency was different [46,54]. The oil-water mixture separation efficiency of the rough morphology of the paraboloid was higher than that of the oil-water mixture with the rough morphology of the truncated cone, demonstrating that superhydrophobicity and oleophilicity facilitate the separation of mixtures of oil and water [54,61]. Moreover, taking the benzene-water mixture as an example, after five tests using the mixture of benzene and water, it was found that the oil-water separation efficiency of the parabolic morphologies was 96% ± 2% and the oil-water separation efficiency of the truncated cone morphologies was 93% ± 1%.

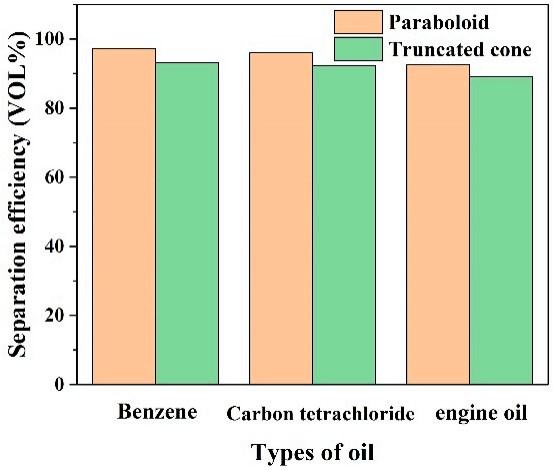

**Figure 10.** Results of separation efficiency for different oil-water mixtures.

*3.6. The Height of Pressure Resistance Study*

The height of the pressure resistance of the meshes has a great influence on oil-water separation. For the same membranes, the higher the level of the liquid, the greater the pressure is and the faster the separation rate, though separation quality may decline a little. When the pressure reaches a critical state, the water will pass through. In order to improve the separation rate, it is required that the membranes are used at the deepest height of pressure resistance as much as possible. A device diagram of highest pressure resistance is shown in Figure 11a. The deionized water was slowly added from the upper of the glass tube along the wall of the glass tube. As the height of the liquid column increased, the deionized water was slowly added using a plastic dropper. When the first water droplet flowed into the segregation apparatus, the height of the liquid column at that moment was recorded. The above experiment was repeated several times to find a stable value of the height of the pressure resistance.

The height of the pressure resistance pattern of the paraboloid and truncated cone morphologies are shown in Figure 11b,c. The height of the pressure resistance value of the copper meshes of paraboloid and truncated cone morphologies was 12 cm of water and 6.9 cm of water, respectively.

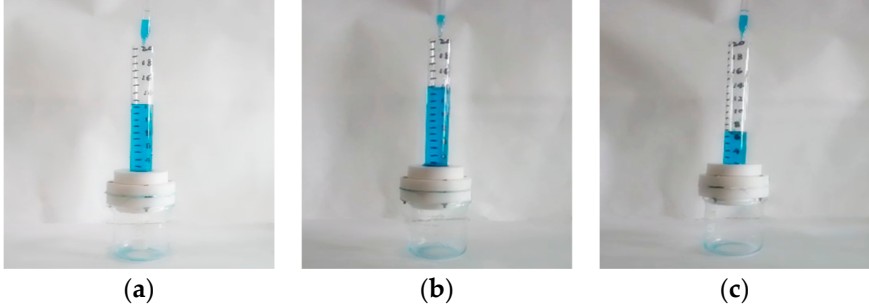

|      |      |      |
| :--: | :--: | :--: |
| (**a**) | (**b**) | (**c**) |

**Figure 11.** (**a**) The device used for highest pressure resistance; (**b**) apparatus of pressure resistance performance test for the parabolic morphology; (**c**) apparatus of pressure resistance performance test for the truncated cone morphology.

To further describe the separation mechanism of oil and water for the SA-modified meshes, SEM was used to observe the mesh membranes with morphologies of the paraboloid and truncated cone, as shown in Figure 12. Jiang et al. [62] have showed that the deepest height of pressure resistance value of superhydrophobic and superlipophilic membranes by capillary phenomenon, the deepest height of pressure resistance, *h*, may be calculated using Equation (2), which is

$$h = -2\gamma_{H_2O}\cos\theta/\rho g R \qquad (2)$$

where $\gamma_{H_2O}$ is the interfacial tension of liquid, $\gamma_{H_2O} = 72.75$ mN/m, $\theta$ represents the water and membrane CA, $\rho_{H_2O}$ is the density of water (1 g/cm$^3$), g represents gravitational acceleration (9.8 kg/N), and *R* is the diagonal length of rectangular meshes.

Copper meshes with rough surfaces which have paraboloid and truncated cone morphologies are shown in Figure 12. In Figure 12a, the copper meshes were covered with a layer of SA. Therefore, the mesh hole was roughly rectangular and the size of the mesh hole was on average about 58 μm × 23 μm; the diagonal length *R* of the rectangle mesh hole was about 62 μm. The deepest height of pressure resistance, $h_1$ (cm), was calculated using Equation (2): The deepest height of pressure resistance $h_1$ of the meshes was 21.4 cm of water. In Figure 12b, the mesh hole was roughly square and the size of the mesh hole was on average about 28 μm × 28 μm; the diagonal length *R* of the square mesh hole was about 40 μm. The deepest height of pressure resistance, $h_2$ (cm), was calculated using Equation (2): the deepest height of pressure resistance $h_2$ of the meshes was 19.6 cm of water. It was smaller than $h_1$.

It can be seen that both the theoretical value and the actual calculated value prove that the deepest height of pressure resistance of the parabolic copper mesh was significantly larger than that of the truncated cone meshes, which confirmed that the membranes with a parabolic rough morphology had superior superhydrophobicity and deepest height of pressure resistance compared to those with a truncated cone morphology.

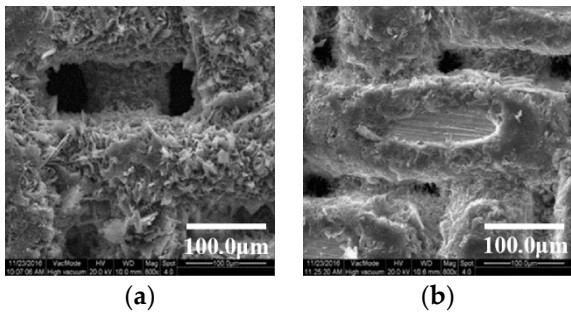

|      |      |
| :--: | :--: |
| (**a**) | (**b**) |

**Figure 12.** SEM of the copper meshes: (**a**) parabolic morphology; (**b**) truncated cone morphology.

In order to clearly illustrate the difference between the heights of pressure resistance of the different morphologies, the results are listed in Table 2.

**Table 2.** The values of the height of pressure resistance under different morphologies.

| Different Situations | Height of Pressure Resistance (cm of Water) | |
| --- | --- | --- |
| | **Parabolic** | **Truncated Cone** |
| Theoretical value | 12 | 6.9 |
| Actual value | 21.4 | 19.6 |

## 4. Conclusions

Two different hydrophobic surfaces on copper meshes were successfully formed by chemical etching, oxidation, and grinding. After modification with SA, surfaces with parabolic morphology exhibited superhydrophobicity. Surfaces with truncated cone morphology showed hydrophobicity.

The WCA and the advancing and receding contact angles of the parabolic morphology were about 153.6°, 154.6° ± 1.1°, and 151.5° ± 1.8°, respectively. The WCA and the advancing and receding contact angles of the truncated cone morphology were about 121.8°, 122.7° ± 1.6°, and 119.6° ± 2.7°, respectively.

The theoretical height of pressure resistance of the parabolic morphology was 12 cm of water and the actual value was 24 cm of water. The theoretical height of pressure resistant of the truncated cone morphology was 6.9 cm of water and the actual value was 19.6 cm of water. This showed that the parabolic morphology had a higher height of pressure resistance than the truncated cone morphology.

The most important thing was to study the separation efficiency of three kinds of water-oil mixtures on the two morphologies. It was found that the parabolic morphology had higher separation efficiency for different oil-water mixtures than the truncated cone morphology. That the parabolic morphology was more superhydrophobic than the truncated cone morphology was confirmed by analyzing the CA, the height of pressure resistance, and separation efficiency. This result illustrates that microstructure is one of the main factors affecting superhydrophobic properties.

**Author Contributions:** Conceptualization, Y.C.; Data Curation, Z.W. and R.S.; Funding Acquisition, S.Y.; Investigation, Q.Z.; Methodology, Q.Z.; Project Administration, S.Y.; Software, C.Y.; Validation, D.Z.; Visualization, R.W.; Writing—Original Draft, Y.C.; Writing—Review and Editing, S.Y., W.W. and Y.Z.

**Funding:** This research was funded by the National Key Research and Development Program of China (No. 2016YFC0400701 and 2018YFC0406504), the National Natural Science Foundation of China (No. 41672224) and Henan province transportation science and technology project (No. 2017J4-1).

**Acknowledgments:** We are highly grateful to the anonymous reviewers for their valuable comments. We also acknowledge the endless support from the staff of Shaanxi Normal University, Xi'an, China.

**Conflicts of Interest:** The authors declare no conflict of interest.

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
