# Peer review of "Effects of Surface Microstructures on Superhydrophobic Properties and Oil-Water Separation Efficiency"

_coatings, doi:10.3390/coatings9020069_

Reviewer 1 Report

This is a review on the paper titled “Effect of Surface Microstructure on Superhydrophobic Properties and Oil-water Separation Efficiency”. In this paper the authors explained the fabrication process of a superhydrophobic copper mesh surface based on etching, oxidizing, grinding and low surface-energy coating. They claim to fabricate two different types of microstructures named: truncated cone and paraboloid. The wetting properties of these materials were characterized. They conclude that the paraboloid geometry works better in respect to separation of oil-water solutions. There is a lack of experiments to confirm the claim. I recommend a major revision before publication. Comments are as follow:

1. The abstract is written more like a summery and not an abstract. Please re-write it and keep the summery at the end of the conclusion section.

2. A schematic for fabrication process is needed for both type of geometries. There is lack of explanation how distinguish fabrication of the different geometry.

3. Line 51: “Therefore, the contact angle is an important index to reveal superhydrophobic properties”

In fact, advancing and receding contact angles are critical to characterize superhydrophobic surfaces. Here you can refer to these references as follow:

  4. Contact-Angle Hysteresis on Super-Hydrophobic Surfaces, DOI: 10.1021/la0486584

5. Robust hybrid elastomer/metal-oxide superhydrophobic surfaces, DOI: 10.1039/C6SM01095D

6. Line 57: Typo mistake: “protrusive surfaces were protrusive surfaces”

7. Line 64: Typo mistake: Marmur

8. Line 74: Silicon can be added to the list:

9.  Robust Superhydrophobic Silicon without a Low Surface-Energy Hydrophobic Coating, DOI: 10.1021/am507584j

10. Line 87: “special materials”, better to call them low surface energy coatings.

11. Line 101: Typo mistake: “rough surface surfaces”

12. Line 103-107, Please break the long sentence down using shorter sentences. Also, there are typo mistakes here and there.

13. Line 112: Please explain in detail all the properties and the providers of each used material.

14. Line 124: The volume ratio of what to what is 1:3?

15. Line 140: Fluorescence microscope is only useful when you have a fluorescent material otherwise there is no point to use and report it. As it is clear in Figure 1, there is no information obtained from the fluorescence microscope images.

16. Line 145: Advancing and receding contact angle measurements are needed for the characterization

17. Line 159-160: “Figure 1b shows that copper meshes etched and oxidized were obviously narrowed.” Not convincing at all. Terms like “obviously” do not have any scientific weights. If you want to be scientifically convincing you should provide data. Do the proper measurements of the width of the wires and make an average of at least 10 measurements on different spots. After collecting the data, you can make a scientific conclusion.

18. Figure 2: The labels should be corrected

19. In Figure 2 and the paragraph related to it, it is not clear if the surface roughness is increased or decreased after SiC treatment. SEM images show less roughness after SiC. Is that the case?

20. Line 198-199: “Therefore, it was very likely that the oil-water separation membranes affected the separation rate because the surface of the copper meshes was stabilized by SA.” How did you conclude that? 

21. Figure 3: why does the SEM of the surface in Fig. 3c is different from Fig. 2d? What is the difference of these two surfaces? Aren’t they the surfaces after etching and oxidation and SiC treatment? Please first clearly explain how the surfaces were fabricated and then explain their properties and characteristics.

22. Line 222 based on Fig.4, “prepared hydrophobic surface and stearic acid, respectively” should be changed to: stearic acid and prepared hydrophobic surface, respectively.

23. Line 251: Advancing and receding contact angles should be presented here.

24. Line 260: “Cao et.al… which showed that SA had better modification ability than DDT.” This is not necessarily correct. The surface contact angle is a function of surface chemistry and surface geometry.

25. Line 268: what is “excellent superhydrophobicity.”? The meaning of superhydrophobic surface is clear but excellent superhydrophobic surface???

26. Line 271: The surface tension of the oils should be reported.

27. Line 297-316 should be re-written with more details. It is not clear what authors are trying to say. The separation experiments should be filmed and presented. There are three different oil-water solutions on two different materials. So, six movies should be presented for this section. The mass of each solution (oil and water) before and after separation should be carefully presented. The method how the measurements has been done should be carefully presented.

28. Line 345: “The height of the pressure resistance of the meshes has a great influence on oil-water separation.” How?

Author Response

Response to Reviewer 1 Comments

Thanks for all your good suggestions and we think it is a really essential and worth considering comment. According to your suggestions and references you provided, we have carefully revised our manuscript, especially include our language, contents, abstract, figures and results. All of the grammatical errors and typo errors you mentioned have been revised. The specific comments mentioned was revised in the following (language errors are not listed):

1. The abstract is written more like a summery and not an abstract. Please re-write it and keep the summery at the end of the conclusion section

According to your proposal, we have rewritten the abstract at line 17-32.

2. A schematic for fabrication process is needed for both type of geometries. There is lack of explanation how distinguish fabrication of the different geometry

We have shown the preparation process of the two morphologies in the form of a schematic diagram, and the differences between the two preparation processes can be clearly seen at line 128-130.

3.  Line 51: “Therefore, the contact angle is an important index to reveal superhydrophobic properties” In fact, advancing and receding contact angles are critical to characterize superhydrophobic surfaces (1: Contact-Angle Hysteresis on Super-Hydrophobic Surfaces, DOI: 10.1021/la0486584  2: Robust hybrid elastomer/metal-oxide superhydrophobic surfaces, DOI: 10.1039/C6SM01095D)

We think the suggestions you provided are really essential. We have carefully learnt the references you have recommended, and have added the advancing and receding contact angles in the introduction at line 45-50.

4. Line 57: Typo mistake: “protrusive surfaces were protrusive surfaces”

    Line 64: Typo mistake: Marmur

    Line 101: Typo mistake: “rough surface surfaces”

We have already modified these typo errors in the paper at line 56, 63 and 99.

5. Line 74: Silicon can be added to the list: Robust Superhydrophobic Silicon without a Low Surface-Energy Hydrophobic Coating, DOI: 10.1021/am507584j

Thank you for the reference you provided. We have carefully learnt the references. We have added silicon in the introduction at line 73, and 80-81.

6. Line 87: “special materials”, better to call them low surface energy coatings.

Line 103-107, Please break the long sentence down using shorter sentences. Also, there are typo mistakes here and there

We have changed“special materials” as “low surface energy coatings” at line 85. We simplified the complex sentences at line 101-104.

7. Line 112: Please explain in detail all the properties and the providers of each used material

We have modified the content in section 2.1, and explained in detail all the properties and the providers of each used material at line 109-115.

8. Line 124: The volume ratio of what to what is 1:3?

The volume ratio of 1:3 refers to the volume ratio of ethanol to water. We have explained this question in the paper at line 123-124.

9. Line 140: Fluorescence microscope is only useful when you have a fluorescent material otherwise there is no point to use and report it. As it is clear in Figure 1, there is no information obtained from the fluorescence microscope images

We have verified this question. The photos were taken by fluorescence microscope. However, the excitation light source of the fluorescence microscope was not turned on when we took photos, and the photos were not fluorescently stained. The photos are actually ordinary photomicrographs. To avoid misunderstanding, we have removed the word “fluorescence” in the paper.

10. Line 145: Advancing and receding contact angle measurements are needed for the characterization

We added the advancing and receding contact angles in section 2.4.2 at line 147-152.

11. Line 159-160: “Figure 1b shows that copper meshes etched and oxidized were obviously narrowed.” Not convincing at all. Terms like “obviously” do not have any scientific weights. If you want to be scientifically convincing you should provide data. Do the proper measurements of the width of the wires and make an average of at least 10 measurements on different spots. After collecting the data, you can make a scientific conclusion

We think the questions you provided are very reasonable. For the sake of rigor, we have used “slightly” to replace “obviously”. Taking into account the scale bars, in order to make it easier to calculate, we have randomly selected 5 points in the SEM image, and measured each point twice. The average diameter was calculated. As a result, it was found that the diameter of the treated copper meshes was slightly narrower than the diameter of the untreated copper meshes at line 163-168.

12. Figure 2: The labels should be corrected

In Figure 2 and the paragraph related to it, it is not clear if the surface roughness is increased or decreased after SiC treatment. SEM images show less roughness after SiC. Is that the case?

We have modified the label of Figure 2. Part of the contents related to Figure 2 have been rewritten. The grinding purpose of SiC is to flatten the sharp portion of the etched and oxidized copper meshes, to make the surface of the copper meshes form a rough structure similar to a truncated cone at line 176-184,189-190 .

13. Line 198-199: “Therefore, it was very likely that the oil-water separation membranes affected the separation rate because the surface of the copper meshes was stabilized by SA.” How did you conclude that?

I am sorry that my mistakes have caused you misunderstanding. I intended to express “separation efficiency”, but I used the wrong words “separation rate”. According to your proposal, I found that this sentence is not very suitable for this position, so I have deleted this sentence of the paper.

14. Figure 3: why does the SEM of the surface in Fig. 3c is different from Fig. 2d? What is the difference of these two surfaces? Aren’t they the surfaces after etching and oxidation and SiC treatment? Please first clearly explain how the surfaces were fabricated and then explain their properties and characteristics.

Based on your suggestions, we have modified some of the expressions in this section to better understand the preparation of these surfaces. Figure 2d shows the preparation process by etching, oxidizing and grinding, and finally a morphology similar to a truncated cone can be obtained. Figure 3c shows the preparation process by etching, oxidizing and modifying with SA, and finally a morphology similar to a parabolic can be obtained at line 176-179,196-201.

15. Line 222 based on Fig.4, “prepared hydrophobic surface and stearic acid, respectively” should be changed to: stearic acid and prepared hydrophobic surface, respectively.

We have changed “prepared hydrophobic surface and stearic acid” as “stearic acid and prepared hydrophobic surface, respectively” at line 227-228.

16. Line 251: Advancing and receding contact angles should be presented here.

Based on your suggestions, we have supplemented the contents of advancing and receding contact angles. We have added advancing and receding contact angles and figures in the paper at line 258-265.

17. Line 260: “Cao et.al… which showed that SA had better modification ability than DDT.” This is not necessarily correct. The surface contact angle is a function of surface chemistry and surface geometry.

We think your opinion is very reasonable. In the preparation of superhydrophobic surfaces, it is important to grasp the surface structure and preparation process. So we have made changes in the paper at line 268-270.

18.  Line 268: what is “excellent superhydrophobicity.”? The meaning of superhydrophobic surface is clear but excellent superhydrophobic surface???

I’m sorry for the inconvenience caused by my mistake. What I want to express is very good superhydrophobic properties. So I have changed “excellent” as “good” in the paper at line 275.

19. Line 271: The surface tension of the oils should be reported.

We have already added the surface tension of the oils at line 279-280.

20. Line 297-316 should be re-written with more details. It is not clear what authors are trying to say. The separation experiments should be filmed and presented. There are three different oil-water solutions on two different materials. So, six movies should be presented for this section. The mass of each solution (oil and water) before and after separation should be carefully presented. The method how the measurements has been done should be carefully presented

According to your suggestion, we have rewritten this paragraph at line 308-331. In the oil-water separation experiment, we have a complete record of the starting point and the end point. The oil-water separation experiments also have videos (as shown in attachment 1). But I am very sorry that the videos are not complete. Based on these records, we supplemented the methods of measurement and data. Thank you for your support of our work.

21. Line 345: “The height of the pressure resistance of the meshes has a great influence on oil-water separation.” How?

For the same membranes, the higher the level of the liquid is, the greater the pressure is, the separation rate is also faster, separation quality may be declined a little. When the pressure reaches a critical state, the water will pass through. In order to improve the separation rate, it is required that the membranes are used at the deepest height of pressure resistance as much as possible at line 361-364.

Reviewer 2 Report

The authors present an interesting study of the effects of surface microstructure in separation membranes manufactured using etched copper meshes and stearic acid. The separation membranes are designed to separate oil and water. Overall, the study is well designed and results are adequately reported to support the manuscript’s conclusions. The only minor recommendations are to improve the grammar (for example, lines 113-115 are not a complete sentence) and abstract. The first sentence or two of the abstract should be more general and introduce the paper and motivation. And the reported pressures in “cm” do not mean anything. I think the authors mean cm of water.

Author Response

Response to Reviewer 2 Comments

Thanks for all your good suggestions and we think it is a really essential and worth considering comment. According to your suggestions you provided, we have carefully revised our manuscript, especially include our grammar, contents and abstract. The specific comments mentioned was revised in the following:

1. The authors present an interesting study of the effects of surface microstructure in separation membranes manufactured using etched copper meshes and stearic acid. The separation membranes are designed to separate oil and water. Overall, the study is well designed and results are adequately reported to support the manuscript’s conclusions. The only minor recommendations are to improve the grammar (for example, lines 113-115 are not a complete sentence) and abstract. The first sentence or two of the abstract should be more general and introduce the paper and motivation. And the reported pressures in “cm” do not mean anything. I think the authors mean cm of water.

Thank you for your comments, we have made changes in the paper at line 11-32, 109-115. For the representation of “cm”, we referred to the expression of reference 63(as shown in the following figures), but what we want to express is water pressure, so we made a revision in the paper for better understanding. We have changed “cm” as “cm of water” in the paper.

Reviewer 3 Report

The submitted manuscript deals with modifications of substrates to obtain hydrophobic and lipophilic materials with high efficiency in oil-water separation. The strength of the presented study is the verification of theoretical prediction through experimental tests. The paper is well organized; a proper description of the state-of-the-art is illustrated; the analytical techniques chosen to investigate the properties of the coatings are appropriate; the bibliographic references are adequate. The following changes are suggested before the publication.

1. Check Ref. 21 and the author cited at line 64.

2. I suggest to summarize the details in section 2.2 using a graphical scheme.

3. In Figures 1, 2, 3, 6, 7, and 11, the scale bars are difficult to read. Please, modify.

4. In Figure 2, letters c and d are missing.

5. An absorption band at 2848 is described on line 224, while a signal at 2852 is reported in Figure 4. Check these data.

6. Line 269: Change “Wettability of water droplets on the surfaces of the mesh membranes” with “Wettability of the surfaces of the mesh membranes: water droplets on”.

   Line 294: Change “Wettability of oil droplets on the surfaces of the mesh membranes” with “Wettability of the surfaces of the mesh membranes: oil droplets on”.

7. To easily follow the discussion, I suggest to list in table(s) the results in sections 3.4 and 3.6.

The main results should be better highlighted in the conclusion section.

8. Check Ref 57, some details is missing.

9. Finally, carefully revise the text. There are many typos. In addition, some sentences are not clear and need to be rephrased. For examples, lines 81-21, lines 113-114, lines 308-309, and lines 354-355.

Author Response

Response to Reviewer 3 Comments

Thanks for all your good suggestions and we think it is a really essential and worth considering comment. We have carefully revised our manuscript, especially include our language, contents, methods, figures and results. The specific comments mentioned was revised in the following:  

1. Check Ref. 21 and the author cited at line 64.

We have revised the mistakes at line 63.

2. I suggest to summarize the details in section 2.2 using a graphical scheme.

We have shown the preparation process of the two morphologies in the form of a schematic diagram, as shown in the following figure.

3. In Figures 1, 2, 3, 6, 7, and 11, the scale bars are difficult to read. Please, modify.

In Figure 2, letters c and d are missing.

We have modified the label of Figure 2, and have modified the scale bars of Figure 1, 2, 3, 6, 7, 11.

4. An absorption band at 2848 is described on line 224, while a signal at 2852 is reported in Figure 4. Check these data.

I’m sorry for the inconvenience caused by my negligence, we have revised the data in the paper at line 229.

5. Line 269: Change “Wettability of water droplets on the surfaces of the mesh membranes” with “Wettability of the surfaces of the mesh membranes: water droplets on”.

   Line 294: Change “Wettability of oil droplets on the surfaces of the mesh membranes” with “Wettability of the surfaces of the mesh membranes: oil droplets on”.

 We have changed “Wettability of water droplets on the surfaces of the mesh membranes” as “Wettability of the surfaces of the mesh membranes: water droplets on” at line 276.

We have changed “Wettability of oil droplets on the surfaces of the mesh membranes” with” as “Wettability of the surfaces of the mesh membranes: oil droplets on” at line 302.

6. To easily follow the discussion, I suggest to list in table(s) the results in sections 3.4 and 3.6.

According to your suggestion, we will list the contents of sections 3.4 and 3.6 in the form of tables, as shown in Table 1 and Table 2.

Table 1 Contact angles of different materials on different morphologies.

Materials             Angle

Parabolic

Truncated   cone

Water

WCA

153.6°

121.8°

θA

154.6±1.1°

151.5±1.8°

θR

122.7±1.6°

119.6±2.7°

Engine oil

32.9°

Carbon tetrachloride

0.1°

0.1°

Table 2 The values of the height of the pressure resistance under different morphologies

Different situations Height of pressure resistance (cm of water)

Parabolic

Truncated   cone

Theoretical value

12

6.9

Actual value

21.4

19.6

7. The main results should be better highlighted in the conclusion section.

According to your suggestion, we have revised the conclusions at line 404-419.

8. Check Ref 57, some details is missing.

We have modified the references at 565-566.

9. Finally, carefully revise the text. There are many typos. In addition, some sentences are not clear and need to be rephrased. For examples, lines 81-21, lines 113-114, lines 308-309, and lines 354-355.

According to your suggestion, we have modified these questions in the paper at line 17-22, 109-115, 308-324, 367-370.

Reviewer 4 Report

1. By addressing the references, the same style should be used, i.e. Page 1, 42; water [5].

2. Page 5, 182; 5000x magnification

3. Page 7, 241; was the thickness of SA determined?

Author Response

Response to Reviewer 4 Comments

Thanks for all your good suggestions and we think it is a really essential and worth considering comment. According to your suggestions you provided, we have carefully revised our manuscript, especially include our language, contents and references. The specific comments mentioned was revised in the following: (language errors are not listed):

1. By addressing the references, the same style should be used, i.e. Page 1, 42; water [5].

We have revised the references at line 431-567.

2. Page 5, 182; 5000x magnification.

We have revised this question at line 186.

3: Page 7, 241; was the thickness of SA determined?

The thickness of SA coating is between 100μm and 300μm.

Round  2

Reviewer 1 Report

Accept in present form

Reviewer 3 Report

The manuscript has been adequately revised. In my opinion, the paper is now suitable for publication.